# TP5: A Novel Therapeutic Approach Targeting Aberrant and Hyperactive CDK5/p25 for the Treatment of Colorectal Carcinoma

**DOI:** 10.3390/ijms241411733

**Published:** 2023-07-21

**Authors:** Niranjana Amin, Herui Wang, Qi Song, Manju Bhaskar, Sharda Prasad Yadav, Mark R. Gilbert, Harish Pant, Emeline Tabouret, Zhengping Zhuang

**Affiliations:** 1National Institute of Neurological Disorders and Stroke, Bethesda, MD 20892, USA; 2Neuro-Oncology Branch, Center for Cancer Research, National Cancer Institute, National Institutes of Health, Bethesda, MD 20892, USA; 3Translational Neuroscience Center, National Institutes of Health, Bethesda, MD 20892, USA; 4Vaxess Technologies, Woburn, MA 02139, USA; 5Institute of NeuroPhysiopathology (INP), National Centre for Scientific Research (CNRS), Aix-Marseille University, 13005 Marseille, France

**Keywords:** colorectal carcinoma (CRC), CDK5, TP5, DNA damage, xenograft mouse model

## Abstract

Colorectal carcinoma (CRC) is a prevalent cancer worldwide with a high mortality rate. Evidence suggests that increased expression of Cyclin-dependent kinase 5 (*CDK5*) contributes to cancer progression, making it a promising target for treatment. This study examined the efficacy of selectively inhibiting CDK5 in colorectal carcinoma using TP5, a small peptide that selectively inhibits the aberrant and hyperactive CDK5/p25 complex while preserving physiological CDK5/p35 functions. We analyzed TP5’s impact on CDK5 activity, cell survival, apoptosis, the cell cycle, DNA damage, ATM phosphorylation, and reactive oxygen species (ROS) signaling in mitochondria, in CRC cell lines, both alone and in combination with chemotherapy. We also assessed TP5’s efficacy on a xenograft mouse model with HCT116 cells. Our results showed that TP5 decreased CDK5 activity, impaired cell viability and colony formation, induced apoptosis, increased DNA damage, and led to the G1 phase arrest of cell cycle progression. In combination with irinotecan, TP5 demonstrated a synergy by leading to the accumulation of DNA damage, increasing the γH2A.X foci number, and inhibiting G2/M arrest induced by Sn38 treatment. TP5 alone or in combination with irinotecan increased mitochondrial ROS levels and inhibited tumor growth, prolonging mouse survival in the CRC xenograft animal model. These results suggest that TP5, either alone or in combination with irinotecan, is a promising therapeutic option for colorectal carcinoma.

## 1. Introduction

Colorectal cancer (CRC) is a widespread disease and one of the most common cancers globally, with approximately 1 million new diagnoses each year [1]. Unfortunately, a significant portion of CRC patients, around 25%, are diagnosed with advanced stages of the disease. Additionally, roughly half of CRC patients will experience the development of metastases, which contributes to the high mortality rates associated with the disease [2,3]. Today, despite recent advances, the median overall survival for patients with metastatic CRC (mCRC) is around 30 months, and five-year survival rates remain less than 15% [4]. Therefore, identifying new treatment options remains a priority for patients with mCRC.

Cyclin-dependent kinase 5 (CDK5) has been increasingly implicated in cancer progression. It is a unique type of cyclin-dependent kinase, predominantly expressed in the brain [5]. Normally, CDK5 is activated by binding to its non-cyclin activators p35 or p39, which are anchored to cell membranes. However, in pathological conditions such as cancer, these cofactors can be cleaved by calpain, producing p25 and p29 with longer half-lives and increased solubility, allowing them to access cytoplasmic and nuclear substrates [6]. CDK5/p25 is the major form in tumorigenesis and has been linked to cell proliferation, cell cycle regulation, cell migration, DNA repair, and drug resistance [7,8,9]. Additionally, CDK5 has been found to be activated in cancer cells or tissues upon exposure to conventional DNA-damaging therapies such as genotoxic agents [10,11].

It was reported that *CDK5* was highly expressed and activated in CRC. In tumor tissues, *CDK5* was overexpressed compared to normal tissue [12]. TCGA analyses revealed that *CDK5* was upregulated in primary CRC due to increased copy numbers [13]. In contrast, *CDK5* was rarely mutated among cancers and, more importantly, was never mutated in CRC, indicating that *CDK5*’s contribution to CRC oncogenesis was due to its high expression rather than mutation [13]. Finally, high *CDK5* expression was reported to correlate with a poor patient outcome: high *CDK5* expression was associated with an advanced stage, poor differentiation, an increased tumor size, and poor patient prognosis [12,14].

The growing evidence of CDK5’s role in cancer progression has made it a potential target in oncology, particularly in the treatment of metastatic colorectal cancer, where genotoxic agents are often used as a first-line therapy. TP5, a small 24-amino-acid peptide developed at the National Institutes of Health (NIH), specifically inhibits the tumor-related CDK5/p25 activity while preserving the normal endogenous CDK5/p35 form and its physiological functions [15,16]. In this study, we investigated the efficacy of TP5 in vitro and in vivo against colorectal cancer. Our results showed that TP5 reduced cancer cell viability and tumor growth by disrupting the cell cycle and increasing DNA damage-induced apoptosis. Additionally, we demonstrated that TP5 enhanced the effectiveness of chemotherapy when used in combination.

## 2. Results

### 2.1. TP5 Inhibits CDK5 Activity and Decreases Cell Viability and Migration

We first evaluated the capacity of TP5 to inhibit the CDK5 activity using immuno-precipitation and a CDK5 activity assay on two CRC cell lines: HCT116 and HT29 cells. We observed that TP5, but not the scrambled control peptide, decreased CDK5 activity in a dose-dependent manner (Figure 1A). We also treated these two cell lines with TPF5, a conjugated form of TP5 with fluorescein isothiocyanate (FITC) at the N-terminus, and observed a ubiquitous FITC signal in the cytoplasm and nuclei of the cells (Figure 1B). These results confirmed that TP5 could efficiently penetrate the cells and inhibit CDK5 activity.

We then tested TP5’s efficacy in vitro using three CRC cell lines: HCT116, HT29, and Caco2 cells. The cell counting kit-8 (CCK-8) assay confirmed that TP5 alone decreased cell viability in a dose-dependent manner (Figure 1C) in all three cell lines, with IC50 concentrations ranging from 16.5 µM to 25.8 µM (Figure 1C). In the clonogenic assay, TP5 significantly decreased the colony number in a dose-dependent manner in both HCT116 and HT29 cells (Figure 1D). Finally, the wound healing assay showed that TP5 at 25 µM significantly reduced the migration capacity of both HCT116 and HT29 cells (Figure 1E).

### 2.2. TP5 Alone Induces Cell Apoptosis by Impairing the Cell Cycle and DNA Damage Repair

To assess the impact of TP5 on cell survival, we analyzed cell apoptosis after TP5 treatment, using flow cytometry staining (Annexin V/PI) and Western blotting. Results showed that TP5 induced both early and late apoptosis, as indicated by increased Annexin V/PI staining (Figure 2A,B). Western blot analysis further confirmed the increase in cleaved PARP and cleaved caspase 3 protein levels in TP5-treated HCT116 and HT29 cells (Figure 2C and Appendix A).

To understand the increased apoptosis in TP5-treated cells, we analyzed DNA damage and cell cycle changes. Our results indicated that TP5 led to a significant, dose-dependent increase in γH2A.X, as determined by both immunofluorescence staining and Western blotting, in HCT116 and HT29 cells (Figure 2C,D; Appendix A). We also observed increased G1 phase% and decreased G2 phase% and S phase% in TP5-treated HCT116 and HT29 cells, indicating that TP5 caused G1 phase arrest in CRC cells (Figure 2E,F). These results suggest that TP5 treatment induces apoptosis by impairing DNA damage repair in CRC cells.

### 2.3. TP5 Acts Synergistically with Chemotherapy by Increasing DNA Damage

One standard of care chemotherapy for colorectal cancer (CRC) patients is irinotecan, which induces potent DNA damage in tumor cells through its active metabolite Sn38, which inhibits topoisomerase-I. Given that TP5 also causes DNA damage in CRC cells, we investigated the potential synergistic effect of TP5 and Sn38. The CDK5 activity assay showed that Sn38 significantly increased CDK5 activity, while combination with TP5 decreased CDK5 activity in both HCT116 and HT29 cells (Figure 3A). Moreover, combination treatment with TP5 and Sn38 significantly increased γH2A.X in both cell lines, suggesting the accumulation of DNA damage (Figure 3B,C). Accordingly, the TP5 and Sn38 combination induced the highest protein levels of cleaved PARP and caspase 3 in both HCT116 and HT29 cells, indicating a synergistic effect of TP5 and Sn38 in DNA damage-induced apoptosis (Figure 3B, Appendix A). We also confirmed the synergistic effects of TP5 and Sn38 using a clonogenic assay. The combination of Sn38 with TP5 caused a dramatic reduction in colony formation compared to single-drug treatment groups (Figure 3D). Finally, adding TP5 to Sn38 also impaired the G2/M arrest induced by Sn38, leading to the accumulation of cells in the G1 phase (Figure 3E,F). These results suggest that TP5 acts synergistically with Sn38 by reducing the ability of tumor cells to repair DNA damage caused by Sn38.

We previously reported that TP5 acts synergistically with temozolomide and irradiation in glioblastoma cells by impairing the DNA damage repair ability in tumor cells by inhibiting ATM phosphorylation [17]. Accordingly, we also checked the p-ATM level in TP5 and/or Sn38-treated CRC cells. We quantified the phosphorylation level of ATM in CRC cells by flow cytometry staining with a PE-conjugated antibody targeting phosphorylated ATM. Sn38 treatment at 20 ng/mL for 24 h significantly increased p-ATM in HT29 and HCT116 cells, indicating that the DNA damage repair machinery was activated in response to Sn38-induced DNA damage. However, TP5 treatment alone or in combination with Sn38 did not reduce p-ATM significantly, suggesting a different mechanism of action of TP5 in CRC cells (Figure 4A,B).

CDK5 also localizes on the inner mitochondria membrane, and loss of CDK5 in breast cancer cells was reported to promote ROS accumulation through dysregulation of the mitochondrial permeability transition pore [18,19]. We then checked whether TP5 treatment also increased mitochondria ROS in CRC cells by MitoSOX staining. Interestingly, TP5 treatment alone increased the mitochondria ROS level in a dose-dependent manner, and combination with Sn38 significantly increased mitochondria ROS further (Figure 4C). These data indicate that TP5 acts synergistically with Sn38, at least partially, by increasing the mitochondria ROS level, leading to more DNA damage in tumor cells.

### 2.4. TP5 Suppresses CRC Tumor Growth In Vivo

To evaluate the anti-tumor efficacy of TP5 in vivo, we established a subcutaneous CRC model with HCT116 cells. We randomized the tumor-bearing mice into four groups: a vehicle control group (H_2_O), TP5 alone (50 mg/kg, every other day), Sn38 alone (5 mg/kg, every other day), and a combination group (TP5 and Sn38). The results showed that TP5 or Sn38 alone inhibited tumor growth compared to the control group, and the tumor growth was the slowest in the combination group (Figure 5A). The mouse survival time until the study endpoint was also significantly prolonged in the combination group compared with the other three groups (Figure 5B). These results suggest that TP5 acted synergistically with Sn38 in inhibiting CRC tumor growth in vivo.

## 3. Discussion

In this study, we demonstrated that TP5 could reduce the activity of CDK5 in colorectal carcinoma (CRC) cells, leading to cell apoptosis. The treatment also caused disruptions in the cell cycle and impaired DNA damage repair. When combined with the chemotherapy agent irinotecan, TP5 showed a synergistic effect by increasing DNA damage accumulation. Results from CRC mouse models showed that TP5 alone or in combination with irinotecan reduced the tumor volume and prolonged mouse survival, indicating that TP5 may be a promising therapeutic option for CRC patients.

Today, there is an urgent need to develop new therapeutic agents for mCRC patients. First-line treatment is usually composed of the association of fluoropyrimidine-based chemotherapy with irinotecan and oxaliplatin. However, these associations do not allow a 5-year survival rate of more than 15% [20]. More recently, the addition of anti-EGFR therapy has improved the overall survival of patients but can be proposed only for patients with RAS wild-type mCRC. Moreover, these therapies can only delay and not prevent relapse [2]. Anti-BRAF therapies failed to reproduce the impressive results observed for melanoma patients, despite the prevalence of BRAF mutation in more than 10% of patients with mCRC [2]. Finally, immunotherapy has recently garnered approval in microsatellite instability (MSI)-high chemotherapy-refractory mCRC patients. However, they represent less than 10% of mCRC patients [21]. Further development is ongoing for microsatellite-stable (MSS) refractory patients, but the first results remain uncertain [4]. Therefore, there is an urgent need to develop new therapeutic strategies for mCRC patients.

Irinotecan is a commonly used treatment for mCRC, both for patients who are fit for intensive treatment, with the aim of reducing the size of their tumors and making them resectable, as well as for those in whom the goal is to control the disease [2,22]. Improving the efficacy of irinotecan has the potential to enhance both the survival and quality of life of mCRC patients [23].

In the context of colorectal carcinoma, CDK5 represents an attractive target for treatment due to its high expression in tumor tissue, involvement in various aspects of oncogenesis, and potential to enhance the effects of DNA-damaging agents [9,13,14,24]. Historically, CDK5 was not considered to be implicated in cell cycle regulation because of its predominant expression in non-dividing neurons. However, recently, CDK5 was reported to phosphorylate the retinoblastoma protein (Rb), promoting cancer cell proliferation [25]. It has also been shown that CDK5 can be activated by cyclin I, which results in anti-apoptotic functions due to the increased expression of BCL-2 family proteins [26]. Moreover, CDK5 was reported to be implicated in the migration and invasion of cancer cells, especially in CRC cells, reinforcing our results on the impact of TP5 on the CRC cell migration [12,13,27].

Studies have shown that CDK5 plays a crucial role in activating the DNA damage response and repair processes. Upon exposure to environmental stressors, such as chemotherapy or radiation, these mechanisms are triggered to maintain genome stability during cell division and enhance cancer cell survival [24,28,29,30]. We observed an increase in CDK5 kinase activity following Sn38 treatment in both HCT116 and HT29 cells (Figure 3A), indicating that CDK5 is among the early proteins activated in CRC cells after exposure to irinotecan. We hypothesize that Sn38 induces DNA damage in tumor cells, triggering the CDK5-mediated DNA damage repair pathway as a mechanism of resistance to repair the Sn38-induced DNA damage. Upon TP5 treatment, CDK5 activity is inhibited, leading to a reduction in the DNA damage repair capacity of the tumor cells.

Research by Ehrlich et al. also reported that the downregulation of CDK5 activity in hepatocellular carcinoma cells hindered the initiation of the DNA damage response, including the G2/M arrest [24]. These results were in line with those we observed after the exposure of CRC cells to TP5 alone or in combination with irinotecan. Finally, Zhuang et al. investigated specifically the role of CDK5 in CRC and found that CDK5 regulated CRC cell proliferation and metastasis [14]. Using the knockdown of *CDK5*, they showed that CDK5 promoted the proliferation ability of CRC cells in vitro. They also showed that the downregulation of *CDK5* deregulated cell cycle progression, demonstrating the cell line arrest at the G1/S phase transition. They also observed that CDK5 promoted metastasis by increasing cell invasion and migration. These results confirm the impact of CDK5 on CRC proliferation and migration and reinforce our results obtained by blocking CDK5 using TP5, highlighting the interest in this therapeutic development.

In conclusion, TP5 has shown promising results as a potential therapy for mCRC, either as a standalone treatment or in combination with chemotherapy. Further research is necessary to fully assess its efficacy before translating to patients in early clinical trials.

## 4. Materials and Methods

### 4.1. Reagents and Compounds

Antibodies of γH2A.X (Ser139; #9718), CDK5 (#2506), β-actin (#3700), cleaved caspase 3 (Asp175; #9664), and cleaved PARP (Asp214; #5625) were purchased from Cell Signaling Technology (Danvers, MA, USA) for Western blotting. Antibodies for the immunofluorescence staining of γH2A.X (#05-636) were purchased from Millipore (Burlington, MA, USA). PE anti-ATM Phospho (Ser1981) antibody (#651204) was purchased from BioLegend (San Diego, CA, USA). Sn38 was purchased from Abcam (ab141108). TP5/TPF5 and the scrambled peptide were synthesized by Genscript (Piscataway, NJ, USA).

### 4.2. Cell Line Preparation

The HT29, HCT116, and Caco2 were purchased from the ATCC (Manassas, VA, USA). They were cultured in DMEM medium plus 10% fetal bovine serum and 1% penicillin/streptomycin. The initial culturing density was 1.5–3 million per T75 flask, and the cells were passaged every three days with a splitting ratio of 1:8–1:10.

### 4.3. Cell Viability Assay

The cell viability assay using the cell counting kit-8 (CCK-8, Dojindo, Japan) was performed according to the manufacturer’s instructions. Briefly, we seeded 1000 cells per well in 96-well tissue culture plates. Following a 12–24-h incubation period, the cells were treated with either the control or TP5 for 72 h. Subsequently, the medium in each well was replaced with 100 µL of fresh medium. Next, 10 µL of CCK-8 reagent was added to each well and they were incubated for 1 h. The results were measured by quantifying the absorbance at 450 nm using the Synergy H1 microplate reader from Agilent BioTek (Santa Clara, CA, USA).

### 4.4. Colony Formation Assay

The clonogenic assay was performed as described previously [17]. Briefly, we seeded 600 cells per well in 6-well tissue culture plates. After a 24-h incubation period, the cells were treated according to the designated groups (control, TP5, Sn38). The medium was replaced every three days to ensure optimal conditions. Following a ten-day incubation period, the colonies were washed with PBS and stained with a 0.5% crystal violet solution in 20% methanol for 15 min. Subsequently, the colonies were washed three times with PBS. Colonies containing more than 20 cells were counted.

### 4.5. CDK5 Activity Assay

Immunoprecipitation and CDK5 kinase activity were performed as previously described [17]. Briefly, Protein A/G PLUS-Agarose (Santa Cruz Biotechnology, Santa Cruz, CA, USA) was washed three times with 1× TBS buffer. Subsequently, it was incubated with CDK5 antibody (Cell Signaling Technology, Danvers, MA, USA, #2506S, 1–2 μg/500 μg of protein lysate) for one hour at room temperature with gentle mixing. Following the washing steps, the CDK5 antibody-conjugated agarose was incubated with the protein lysates of each experimental group overnight at 4 °C with gentle mixing. After washing with TBS buffer, the agarose was resuspended in kinase buffer containing 50 mM MOPS (pH 7.2), 12.5 mM β-glycerol-phosphate, 5 mM MgCl_2_, 5 mM EGTA, and 2 mM EDTA. For the CDK5 kinase assay, 20 μL of the prepared CDK5 immunoprecipitate was incubated with 10 μg of histone H1 and 0.1 mM [γ-32P] ATP in kinase buffer with a total volume of 50 μL. The reaction mixture was then incubated at 30 °C for one hour. To terminate the reaction, Laemmli sample buffer was added, and the mixture was heated at 90 °C for five minutes. The reaction product was subsequently electrophoresed on a 4–20% SDS-PAGE gel. After electrophoresis, the gel was dried, and the dried gel was exposed overnight for autoradiography to detect the radioactive signals.

### 4.6. Western Blotting

The cell lysates were prepared with RIPA buffer, which contained a protease inhibitor cocktail (4693132001, Roche, Basel, Switzerland) and a phosphatase inhibitor cocktail (4906845001, Roche). The protein lysate of each group (5–20 μg) was electrophoresed on Invitrogen NuPAGE Bis-Tris protein gels (4–12%) and then transferred to nitrocellulose membranes using the iBlot2 gel transfer device (ThermoFisher Scientific, Waltham, MA, USA). The membranes were blocked with BSA or non-fat dry milk and then incubated with primary antibodies overnight at 4 °C. Afterward, the membranes were incubated with HRP-conjugated secondary antibodies for 1 h at room temperature. The Western blot bands were detected using an enhanced chemiluminescence substrate (R1004, Kindle Biosciences, Greenwich, CT, USA) and imaged using the KwikQuant Imager (D1001, Kindle Biosciences).

### 4.7. Immunofluorescence Staining

The tumor cells were plated in Falcon Chambered Cell Culture Slides (#354118) and treated as indicated in the main text. After treatments, the cells were washed with PBS and fixed in 2% paraformaldehyde (PFA), followed by 70% ethanol. Subsequently, the cells were blocked using 5% BSA and then incubated with primary antibodies overnight at 4 °C, followed by incubation with fluorescent-conjugated secondary antibodies. Finally, the images were captured using a Zeiss LSM710 confocal microscope (Zeiss, Oberkochen, Germany).

### 4.8. Wound Healing Assay

HCT116 and HT29 cells were seeded into 6-well plates. Once cells reached 100% confluence, “wounds” were created by scraping lines with a 200 μL pipette tip, and then cells were washed three times in a serum-free medium. The “wounds” were observed at 4, 8, 24, and 48 h and photographed using an EVOS Cell Imaging System (EVOS XL Core Cell Imaging System, Thermo Fisher Scientific). The distances between the two edges of the scratch (wound width) were measured at three sites for each image using the ImageJ v1.53t. The migratory distances were calculated by subtracting the wound width at each time point from the wound width at the time zero point.

### 4.9. Apoptosis Assay

One million cells of each group were stained with Annexin V-APC/PI (BD Biosciences, 550474, Franklin Lakes, NJ, USA). After indicated treatments, the tumor cells were harvested with trypsin/EDTA solution, washed with PBS and Annexin V binding buffer (BD Biosciences, 556454), and then incubated with Annexin V-APC antibody in 100 µL of Annexin V binding buffer at room temperature for 10 min. Following this, 400 µL of Annexin V binding buffer containing 1 µL of propidium iodide (PI, 1 mg/mL) was added to the cells. After 15-min incubation on ice, the stained cells were analyzed within one hour using flow cytometry (BD LSRFortessa SORP). The flow cytometry data were analyzed with FlowJo version 10.9.0, and the gating strategy for early and late apoptotic cells is presented in Appendix A.

### 4.10. Cell Cycle Analysis

For cell cycle analysis, one million cells were washed twice with PBS and fixed in 70% ethanol. Subsequently, the cells were incubated with 500 µL of FxCycle PI/RNase Staining Solution (Thermo Fisher Scientific) at room temperature for 30 min and analyzed by flow cytometry (BD LSRFortessa SORP). The cell cycle phases (G1 phase, S phase, G2 phase) were quantified by the “Cell Cycle” analysis tool in the FlowJo software (FlowJo version 10.9.0).

### 4.11. MitoSOX Staining

Cells were treated in indicated groups for 24 h before collection. Resuspended cells were incubated with 5 μM MitoSox (Thermo Scientific) for 30 min at 37 °C in a 5% CO_2_ incubator. All assays were analyzed using BD SORP Analyzer (BD Biosciences, Franklin Lakes, NJ, USA).

### 4.12. Subcutaneous CRC Model and Treatment

The animal procedures described in this study were carried out by staff affiliated with NCI-CCR and were approved by the NCI Animal Care and Use Committee (ACUC) in accordance with federal regulatory requirements and standards. The intramural NIH Animal Care and Use (ACU) program, including all its components, has been accredited by AAALAC International. Briefly, 1 million HCT116 cells per mouse were injected subcutaneously into the right flanks of NSG mice. We randomized the tumor-bearing mice into the indicated groups when the mean tumor volumes reached 100 mm^3^. Mice were divided into four groups: control (H_2_O, I.P. 150 µL), TP5 alone (I.P. 50 mg/kg every two days), Sn38 alone (I.P. 5 mg/kg every two days), and TP5 and Sn38 combination. Tumor volumes were assessed every other day by standard caliper measurement (V = L × W^2^/2). The survival endpoint was defined as either of the following criteria: (1) tumor volume reached 2000 mm^3^, (2) tumor diameter reached 20 mm.

### 4.13. Statistical Analysis

Statistical analysis was performed using the GraphPad Prism software (version 6.05) and SPSS v22. Significance was defined as *, *p* < 0.05; **, *p* < 0.01; ***, *p* < 0.001. Graph figures were presented as means with SEM. Statistical analysis was performed using one-way ANOVA and Student’s *t*-test. Mouse survival was analyzed using the Kaplan–Meier and log-rank tests.

## Figures and Tables

**Figure 1 ijms-24-11733-f001:**
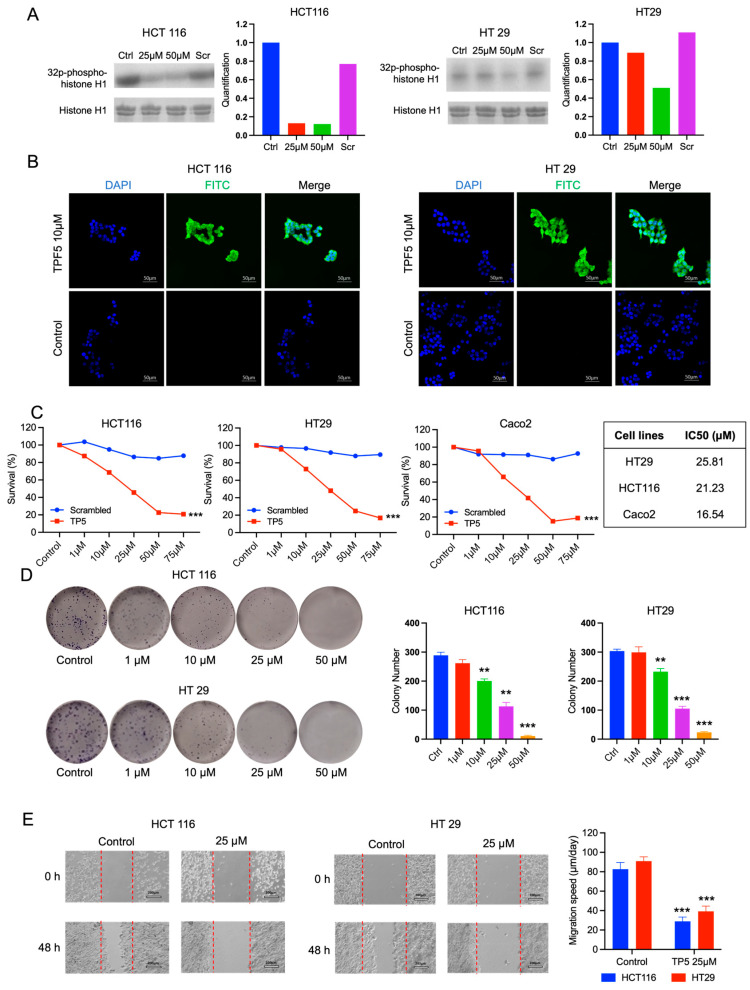
TP5 inhibits CRC tumor cell growth and migration. (**A**) CDK5 activity is indicated by phosphorylated histone 1 in HCT116 and HT29 cells treated by TP5 or scrambled peptide at indicated concentrations. Quantification of the blots is presented on the right side. Ctrl: control; Scr, scrambled control peptide. (**B**) TPF5 (10 µM) efficiently penetrated HCT116 and HT 29 cells. (**C**) The viability of HCT116, HT29, and Caco2 cells treated by indicated TP5 or scrambled peptide concentrations for 72 h. Right table: IC50 concentrations. *** *p* < 0.001. (**D**) Clonogenic growth of HCT116 (top panels) and HT29 (bottom panels) cells. The bar graphs present the quantification of colonies under treatment at indicated concentrations (N = 3). (** *p* < 0.01; *** *p* < 0.001). (**E**) Wound healing assay of HCT116 and HT29 after 48 h of treatment. The bar graphs display the migration speed of the cells (N = 3). (*** *p* < 0.001).

**Figure 2 ijms-24-11733-f002:**
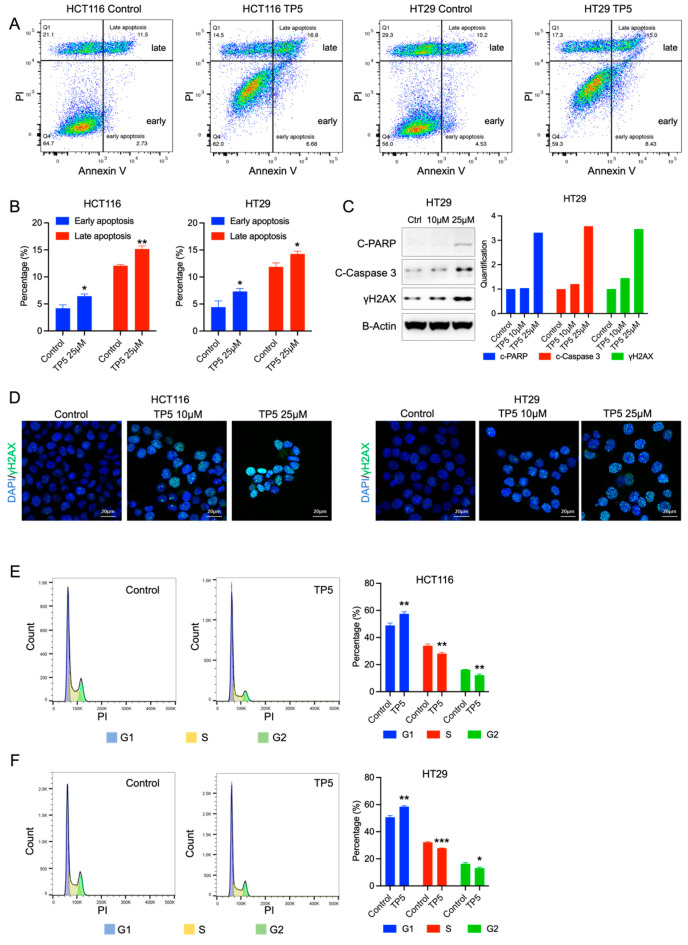
TP5 increases apoptosis and DNA damage and impairs the cell cycle. (**A**,**B**) Apoptosis analysis by Annexin V/PI staining. HCT116 and HT29 cells were treated with TP5 (25 µM) for 24 h and then stained with Annexin V-APC/PI for flow cytometric analysis. (**A**) Representative flow cytometry dot plots using Annexin V-APC/PI staining for apoptosis. Early (Annexin V+ PI−) and late (Annexin V+ PI+) apoptotic cells are labeled in the graphs. (**B**) The percentage of early and late apoptotic cells between the control and TP5-treated cells was statistically compared (* *p* < 0.05; ** *p* < 0.01). (**C**) The protein levels of cleaved PARP, cleaved caspase 3, and γH2A.X in HT29 cells after 24 h of treatment by TP5 (10 µM, 25 µM). Quantification of the blots is presented on the right side. (**D**) Immunofluorescent staining of γH2A.X (green) in HCT116 and HT29 cells after 24 h of treatment by TP5 at indicated concentrations. (**E**,**F**) Cell cycle analysis is shown for HCT116 cells (**E**) and HT29 cells (**F**) treated by TP5 (25 µM) for 24 h. The graph bar displays the cell cycle phase quantification (N = 4; * *p* < 0.05; ** *p* < 0.01; *** *p* < 0.001).

**Figure 3 ijms-24-11733-f003:**
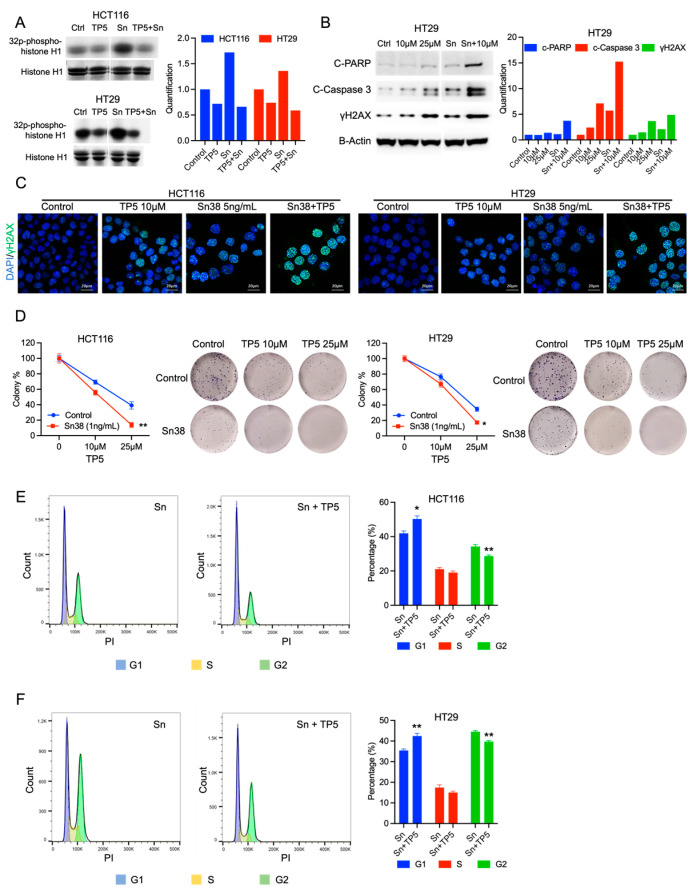
TP5 acts synergistically with Sn38. (**A**) CDK5 activity is shown by phosphorylated histone 1 in HCT116 and HT29 cells treated as indicated (Ctrl: control; Sn: Sn38 5 ng/mL; TP5: 10 µM for HCT116 cells and 25 µM for HT29 cells) for 24 h. Quantification of the blots is presented on the right side. (**B**) The protein levels of cleaved PARP, cleaved caspase 3, and γH2A.X are shown by Western blot in HT29 cells treated as indicated (Ctrl: control; Sn: Sn38 5 ng/mL; TP5: 10 or 25 µM) for 48 h. Quantification of the blots is presented on the right side. (**C**) Immunofluorescent staining of γH2A.X (green) in HCT116 and HT29 cells after indicated treatments for 24 h. (**D**) Synergistic interactions between TP5 and Sn38 are shown by clonogenic growth of HCT116 and HT29 cells treated as indicated (TP5 at 10 and 25 µM) (N = 3; * *p* < 0.05; ** *p* < 0.01) for three days. (**E**,**F**) Cell cycle analysis is shown for HCT116 (**E**) and HT29 (**F**) cells treated by TP5 (25 µM) and Sn38 (5 ng/mL) for 24 h. The graph bar displays the cell cycle phase quantification (N = 4; * *p* < 0.05; ** *p* < 0.01).

**Figure 4 ijms-24-11733-f004:**
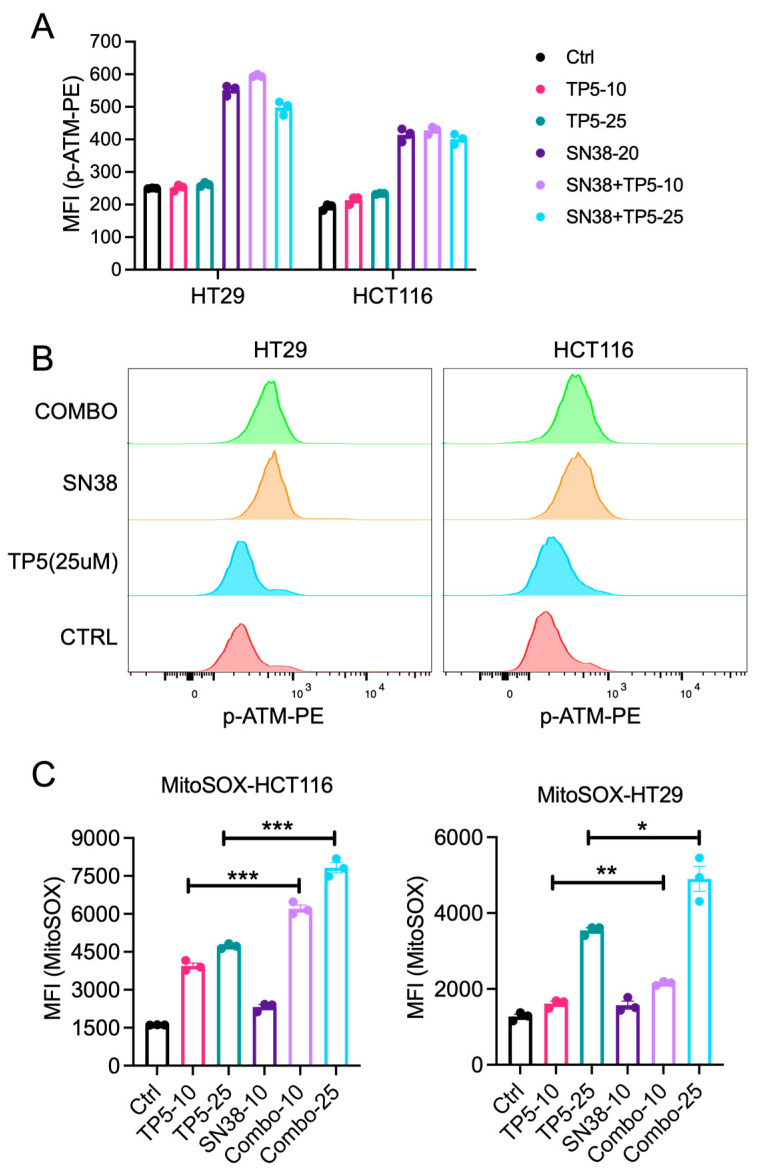
TP5 increased ROS production in CRC cells. (**A**,**B**) p-ATM staining of HCT116 and HT29 cells after 24 h of treatment as indicated. TP5: 10 and 25 µM, Sn38: 20 ng/mL. (**A**) Median fluorescence intensity (MFI) of flow cytometry staining with phosphorylated ATM antibody conjugated with PE (p-ATM-PE) in HCT 116 and HT 29 cells treated as indicated. (**B**) Offset overlay of representative histograms of each treatment group in both HCT116 and HT29 cells. (**C**) MFI of MitoSOX staining in HCT116 and HT29 cells treated as indicated for 24 h (TP5: 10 and 25 µM, Sn38: 10 ng/mL). * *p* < 0.05; ** *p* < 0.01; *** *p* < 0.001.

**Figure 5 ijms-24-11733-f005:**
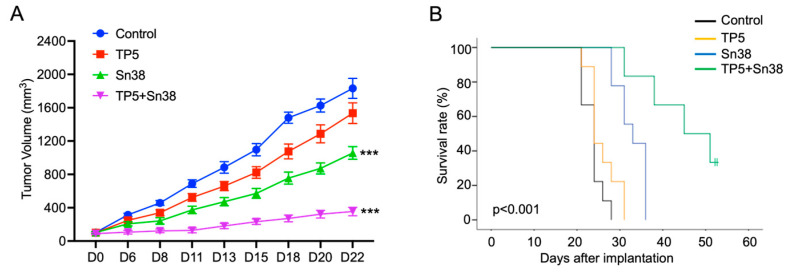
TP5 suppresses CRC tumor growth in vivo. (**A**) Tumor volume over time is shown for the four groups (Control N = 9; TP5 N = 9; Sn38 N = 9; TP5 + Sn38 N = 6) (*** *p* < 0.001). (**B**) Mouse survival time until the study endpoint according to the treatment groups is shown.

## Data Availability

The data that support the findings of this study are available on request from the corresponding authors.

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
