# Peer review of "TP5: A Novel Therapeutic Approach Targeting Aberrant and Hyperactive CDK5/p25 for the Treatment of Colorectal Carcinoma"

_ijms, 2023, doi:10.3390/ijms241411733_

Round 1

Reviewer 1 Report

In the manuscript entitled “TP5: a novel therapeutic approach targeting aberrant and hyperactive CDK5/p25 for the treatment of colorectal carcinoma”, the authors investigate the role of the TP5 peptide in reducing the carcinogenic activity of CDK5, and its potential to treat colorectal carcinoma through induction DNA damage and apoptosis.

The data is well presented and the subject has not been addressed elsewhere. The data is highly relevant, since CDK5 expression have been established as a possible target for several tumors due to its tumor progression activity, including colorectal cancer. In addition, production of TP5 was recently patented (2022) and tested against CDK5 in a glioma model. Thus, TP5 is presented as a novel marker for malignancy and a target for therapy.

The manuscript describes for the first time the mode of action of TP5 in colorectal cancer, its effect in combination with irinotecan in colon cancer and the reduction of colon tumor mass in vivo.

The references are appropriate.

The authors conclusions that TP5 alone or in combination with irinotecan induced ROS, apoptosis, reduced cell growth and migration, and its potential as target to treat colon cancer are supported by the data presented.

There are some variations in font size of the figures that may be improved.

However, I have just a minor concern: Even though I understand that the authors want to reduce space, I believe that not presenting the complete material and methods is deleterious to the paper quality.

Thus, this reviewer suggests the inclusion of the complete material and methods and a careful revision of the font sizes of the figures.

Reviewer 2 Report

The manuscript by Amin et al. describes the anti-tumor activity of TP5 that inhibits CDK5 activity. The authors reported that TP5 treatment induced G1 arrest, apoptosis, DNA damage and increase in mitochondrial ROS levels, and inhibited cell migration in colorectal cancer cells. Cooperation of TP5 with an irinotecan metabolite sn38 was also demonstrated. Therapeutic application of the peptide was explored with tumor xenograft of a colorectal cancer cell. Overall, the subject and results are interesting enough to draw attention of peers. However, design of some experiments and the way of presenting results appear to undermine the quality of the manuscript, which necessitates to be revised significantly.

1.      It is better to give a section number of each section in ‘Results’ and ‘Materials and Methods’ like 2.1. 2. 2. ∙∙∙∙∙.

2.      Line 30. ‘and inhibiting G2/M arrest.’ is confusing. It should be ‘and inhibiting G2/M arrest induced by sn38 treatment.’

3.      Graphs in Figures 1C, 1D, 1E, 2B, 2E, 2F, 3D, 3E, 3F and 5A. The axes and their labels are not easily recognizable. They should be as clear as in Figure 4C. In addition, numbers in y-axis of Figure 2B should be provided to show significant figures.

4.      The legends for Figures 2 and 3 are not detailed enough to tell the treatment conditions, especially the duration of treatment.

5.      Data shown in Figure 2A is not clear enough to show early and late apoptosis. Majority of cells directly become Annexin 5 positive/PI-positive. Cells in Q4 (Annexin 5 positive/PI-negative) are not readily distinguishable. There seems a compensation problem in the experiment. It should be attempted to delineate cells in Q4 by modification of compensation parameters.

6.      Western blot results in Figure 2C and Figure 3B do not match with the original data provided. Were the results of 50 mM TP5 treatment intentionally deleted? TP5 treatment at 50 mM did decrease cell viability and colony formation in Figure 2C and 2D, but did not increase the level of cleaved PARP, cleaved caspase-3 and gH2A.X. Please provide criteria for the deletion.  

7.      What is the criteria to use scrambled peptide instead control? Why was the scrambled peptide used in only a few selected experiments? What is the sequence of the scrambled peptide?

8.      Were the cell cycle analyses in Figures 2E, 2F, 3E and 3F done together? If not, it is required to include cell cycle analysis with control group in Figures 3E and 3F to demonstrate that sn38 treatment induced G2/M arrest.

9.      All of the western blot results and CDK5 activity measurements are required to be analyzed quantitatively.

10.  What is the criterion for following notion at lines 259-260: ‘Our findings also showed that CDK5 is among the early proteins activated in CRC cells after exposure to irinotecan.’  

11.  Treatment of sn38 increased CDK5 activity and co-treatment of TP5 decreased the sn38-induced CDK5 activity to level comparable to TP5 treatment alone. If CDK5 promotes cell survival as predicted from the results, TP5 seems to knock down the resistance capacity of cells against sn38 toxicity. Could it be a mechanism of cooperative action of TP5 and sn38?

12.  Cell culture condition should be described in more detail including splitting ratio and frequency of subculture.

13.  Western blotting methods in reference 17 just refers another reference. The original reference that described the method in detail should be provided here.  

Round 2

Reviewer 2 Report

The authors addressed most of the issues raised in the previous review properly. I appreciate their efforts.